# Synthesis and Characterization of Newly Designed and Highly Solvatochromic Double Squaraine Dye for Sensitive and Selective Recognition towards Cu^2+^

**DOI:** 10.3390/molecules27196578

**Published:** 2022-10-04

**Authors:** Linjun Tang, Shubham Sharma, Shyam S. Pandey

**Affiliations:** Graduate School of Life Science and Systems Engineering, Kyushu Institute of Technology, 2-4 Hibikino, Wakamatsu, Kitakyushu 808-0196, Japan

**Keywords:** double-squaraine dye, solvatochromism, metal ion sensing, colorimetric detection, copper ion chemosensor

## Abstract

Synthesis and characterization of a novel and zwitterionic double squaraine dye (DSQ) with a unique D-A-A-D structure is being reported. Contrary to the conventional mono and bis-squaraine dyes with D-A-D and D-A-D-A molecular frameworks reported so far, DSQ dye demonstrated strong solvatochromism allowing for the multiple ion sensing using a single probe by judicious selection of the suitable solvent system. The DSQ dye exhibited a large solvatochromic shift of about 200 nm with color changes from the visible to NIR region with metal ion sensitivity. Utilization of a binary solvent consisted of dimethylformamide and acetonitrile (1:99, *v*/*v*), highly selective detection of Cu^2+^ ions with the linearity range from 50 μM to 1 nM and a detection limit of 6.5 × 10^−10^ M has been successfully demonstrated. Results of the Benesi–Hildebrand and Jobs plot analysis revealed that DSQ and Cu^2+^ ions interact in the 2:1 molecular stoichiometry with appreciably good association constant of 2.32 × 10^4^ M^−1^. Considering the allowed limit of Cu^2+^ ions intake by human body as recommended by WHO to be 30 μM, the proposed dye can be conveniently used for the simple and naked eye colorimetric monitoring of the drinking water quality.

## 1. Introduction

With the progress of industrial civilization, the water in the environment and the soil are polluted by numerous heavy metals in many ways. This threatens the existence of other species living in these bodies. Currently, the environmental contamination caused by the presence of the toxic metal ions is rising necessitating a significant need for the metal ion sensing. Furthermore, the human body has a large number of metal ions, and their imbalances result in altered body function, leading to several physiological disorders [1]. The quest for the development of novel functional dyes for targeted applications is a never-ending journey of the curious human mind. Especially, the bright tunable colors, synthetic diversity of organic dyes, sensitivity in the near-infrared (NIR) wavelength region, and imparting of the targeted functionality by judicious molecular design has opened the door for applications, particularly in the area of optoelectronics such as optical data storage, imaging, and optical communication [2,3,4,5,6,7]. Amongst functional organic dyes, squaraine dyes, typically bearing a donor(D)-acceptor(A)-donor(D) zwitterionic framework, are one of the most interesting classes of organic dyes owing to their exceptional light absorption [8,9]. The journey of the squaraine dyes started with the report on the synthesis of squaraine dye by condensation of electron-deficient squaric acid with electron-donating pyrrole units by Triebs and Jacob [10]. This geared the momentum of research for the development of such a class of intensely colored dyes utilizing vastly available electron-donating aromatic rings and squaric acid [11,12,13,14]. This quest for the design and development of novel squaraine dyes started about six decades ago, which is expected to be continued in future owing to the plethora of technological applications in the area of probes for bioimaging [15,16], semiconducting elements for organic field-effect transistors [4,5], next-generation solar cells such as organic thin film, dye-sensitized and perovskite solar cells [17,18], huge circular dichroism [19], and fluorescence based electrofluorochromic devices [20,21] along with diverse applications in sensing assisted by molecular self-assembly controlled guest-host interactions [22,23,24,25,26].

Apart from huge synthetic diversity owing to the possibility of molecular engineering in the D-A-D parts of the molecular framework, its beauty lies in the further structural control by designing bis-squaraine and polysquaraine dyes aiming toward tuning the optical absorption window from visible to NIR wavelength regions. At the same time, controlling the optoelectronic properties assisted by the molecular self-assembly led to diverse applications, especially in sensing [27,28,29,30,31]. A novel bis-squaraine type dye, where a flexible multi-glycol unit was flanked by two terminal squaraine dyes was synthesized and reported by Ajayghosh et al. having a calcium ion sensing by H-aggregate type foldamer formation [32]. Molecular self-assembly is an inherent property of squaraine dyes leading to the blue-shifted H-aggregates and red-shifted J-aggregates due to extended π-conjugation due their planer molecular framework [33]. This allows the more enhanced tendency of the molecular self-assembly in the bis-squaraine dyes. Recently it was reported that there is formation of H-aggregate assisted nano-rods and J-aggregate assisted nanosheet-like polymorphs in the newly proposed bis-squaraine dyes covering a large spectral range from 550 nm to 875 nm [34]. Contrary to classical dyes, solvatochromic dyes exhibit change in the optical properties as a function of the changing molecular environment making them to behave as smart molecules having tremendous potential as sensing probes. In this context, orthodox Reichardt’s dye (Betaine 30) showed strong solvatochromism exhibiting a change in color in the whole visible spectrum as a function of changing solvent polarity and ignited the research on the development of solvatochromic dyes for sensing applications [35,36,37]. At the same time, strong color changes in different solvent media allow multi-model detection of target species in different spectral regions using a single probe by the judicious selection of the solvents. Unfortunately, the majority of the organic solvatochromic dyes reported in the literature exhibit optical responses mainly in the visible region of the spectrum. Therefore, there is a need to design and develop novel functional molecules with strong solvatochromic shift from the visible to the NIR wavelength region to enhance the horizon of their potential applications especially in sensing.

Herein, we report the synthesis of a novel double squaraine (DSQ) dye with chemical structure shown in the Figure 1. This dye not only possesses an excellent solubility in a variety of polar and non-polar organic solvents but also strong solvatochromism, paving the way for differential and multi-ion sensing using the same target probe by the judicious selection of a suitable solvent or solvent mixture. Our newly proposed double squaraine dye bears a D-A-A-D double zwitterionic molecular framework. To the best of our knowledge such a structure with a large solvatochromic shift of about 200 nm is being reported for the first time. This newly proposed DSQ dye is different from the previously reported D-A-D structure by typical mono squaraine dyes and D-A-D-A double zwitterionic structure by bis-squaraine dyes. This dye bears alkyl substituted benzo[e]indole as electron-rich donor and squaric acid as electron-deficient acceptor units. To have a control on the molecular self-assembly/dye aggregation, which is a typical behavior of squaraine dyes, alkyl chains at both of the n-positions of the benzo[e]indole ring can be easily introduced and altered to impart the structural and functional diversities and in this present work, we have used the ethyl group as a representative alkyl substituent.

## 2. Results and Discussion

### 2.1. Synthesis of the Double-Squaraine (DSQ) Dye (***1***)

The DSQ dye (**1**) was successfully synthesized in four steps as shown in Figure 1. The key precursor (**5**) was synthesized in three steps from the commercially available starting compound 1,1,2-trimethyl-1*H*-benz[e]indole and diethyl squarate. The reaction intermediate (**3**) was conveniently synthesized in good yield by the quaternization of 1,1,2-trimethyl-1*H*-benz[e]indole (**2**) using iodoethane (**7**) under reflux in acetonitrile. Semi-squaraine dye intermediate (**4**) was prepared by the condensation of (**3**) and 3,4-diethoxysquarate (**8**) using triethylamine as a base to enhance the deprotonation and reactivity of the benzoindolium intermediates (**4**), which was then hydrolyzed to give semisquaraine dye (**5**). The target DSQ dye (**1**) was finally prepared by the acid-catalyzed self-condensation reaction of (**5**) in the presence of an excess thionyl chloride (SOCl_2_) as an acid catalyst in ether. The structural of the final DSQ dye (**1**) and their corresponding intermediates were verified by Mass and NMR spectral investigations (Appendix A). It is worth to mention here that the overall yield of the target dye by this synthesis route is too low (8.2%). In order to improve the synthetic yield of the double squaraine dye, we adopted an improved synthetic route as shown in Figure 2. In this improved method, synthetic intermediate (**3**) was reacted with 0.45 equivalent of squaric acid (**9**) in toluene and 1-butanol mixture (1:1, *v*/*v*) to give the symmetrical mono squaraine dye (**6**). This symmetrical squaraine dye has a double bond, which can be attacked again by nucleophilic squaric acid (**9**), since the reactivity of the double bond is reduced due to the presence of squaric acid (**9**) moiety but reaction conditions, especially the choice of solvent, play a crucial role in the promotion of the condensation reaction. The use of a mixture of toluene and 1-butanol in (5:1, *v*/*v*) ratio was important to obtain the target DSQ dye (**1**) at a 25% yield. This proposed improved method not only improved the overall yield (19%) but also further verified the structure of the DSQ dye (**1**). At the same time, conversion of the typical mono squaraine dye to double squaraine dye by this improved synthetic route opens the door for the synthesis of a variety of symmetrical and unsymmetrical double squaraine dyes by altering the diverse variety of electron-donating heterocyclic units.

### 2.2. Solvatochromism in the DSQ Dye

Solvatochromism is basically a change in the color of target molecule by change in the molecular environment created by solvent molecules by differential interactions between target and solvent molecules. This imparts the inherent tendency of sensing to the solvatochromic molecules by the changing molecular environment as external stimulus and the possibility of controlling this interaction by judicious selection of the suitable solvent media. To investigate the solvatochromism, the synthesized DSQ dye was dissolved in different polar and non-polar organic solvents followed by the observation of change in color by naked eye and spectral investigation by the UV-visible-NIR electronic absorption spectroscopy as shown in the Figure 2. It can be seen from the Figure 2a that, under natural light conditions, there was a distinct change in color by the naked eye for the 20 μM solution of DSQ dye in different polar and non-polar organic solvents demonstrating strong solvatochromism. To investigate this qualitative color change observed by naked eye in more detail, electronic absorption spectra of these different solvents was also recorded and shown in Figure 2b. It can be seen from this figure that DSQ dye exhibits strong solvatochromism having absorption spectral features ranging from visible to NIR wavelength regions (450–800 nm) in different solvents. Contrary to this dye, its mono-squaraine dye (**6**) as well as semi-squaraine dye (**4**) counterparts exhibits weak solvatochromism with only a small change of 20–30 nm in the absorption maximum (λ_max_) (Appendix A) demonstrating that our newly proposed double squaraine dye (DSQ dye) functions as a potential solvatochromic dye probe.

A perusal of the Figure 2b clearly corroborates that the spectral features can be grouped into two major classes, one exhibiting dominant absorption between 600–800 nm by typical dipolar solvents (DMF, DMSO) and protic solvents (methanol, ethanol) while second where the dominant absorption manly in the 450–650 nm exhibited by acetonitrile, tetrahydrofuran (THF), ethyl acetate, toluene, chloroform, acetone, 2-propanol, n-butanol and acetic acid (AcOH). Blue-shift absorption in the 450–650 nm range is thought to be caused by the production of H-aggregates. It has been widely reported that squaraine dyes exhibit blue-shifted spectral features due to H-aggregate formation various efforts has been directed to control this aggregation behavior for controlling the functionality in photovoltaics and sensing [38]. In DMF, DMSO, methanol, ethanol, DSQ dye (**1**) predominantly exists in monomeric state due to these dipolar solvents have significant dye-solvent dipole-dipole interactions to hampering of the dye-aggregation in the monomeric state. The differential of the spectral features of the DSQ dye in protic solvent of methanol, ethanol, *n*-butanol, 2-propanol and AcOH could be attributed to the different of their dielectric constant. This argument is further validated by the fact that contrary to methanol (ε = 32.2) and ethanol (ε = 24.5), use of 2-propanol (ε = 19.9), 1-butanol (ε = 17.5) and AcOH (ε = 6.2) leads to not only visual change in the color of the solution but also pronounced decrease in the absorbance around λ_max_ of 710 nm and concomitant increase in the absorbance around λ_max_ of 580 nm [39].

Considering the two dipolar organic solvents acetonitrile (ACN) and DMF with high and nearly similar dielectric constants [40] of 37.5 and 37.8 exhibit contrasting color and absorption spectral features with λ_max_ of 592 nm and 728 nm, respectively. We investigated the spectral changes of the DSQ dye in the different ratio of binary solvent mixtures consisted of ACN and DMF and results are shown in Figure 2c. The excited-state interactions such as change in the conformation, charge (electron/proton) transfer, and non-covalent interactions with solvent molecules such as van der Waals forces have been widely used to explain differential solvatochromic behavior in the functional organic molecules [41,42,43,44,45,46]. This contrasting solvatochromic behavior in these solvents can be explained considering the differential aggregation behavior of the DSQ dye taking dipole-dipole, and hydrogen bonding interactions between the dye and solvent molecules. DMF is a strong hydrogen bond acceptor (HBA), while acetonitrile has a weak HBA tendency [47]. The strong HBA tendency of DMF favors the dye to exist in the monomeric state. On the other hand, weak HBA of ACN leads to the formation of H-aggregates resulting in the hypsochromic shift in the absorption spectrum. In order to further verify the H-aggregation formation in the DSQ solution, we investigated the absorption spectra of the different concentrations of DSQ dye in DMF (Appendix A). At the lower concentration, the peak at 726 nm appears, indicating the absorption of the dye in mono-molecule state. As the concentration of the DSQ dye solution increases, 554 nm and 600 nm peaks appear, suggesting a blue shifted absorption spectral features. These spectral changes at higher concentration validates the formation of H-aggregation. Kim et al. have also emphasized that squaraine dyes easily form H-aggregates leading to blue-shifted absorption spectral features [48]. Another possible reason for this contrasting spectral behavior between the ACN and DMF could be attributed to the differential molecular planarization in these solvents. The possibility of strong coordination between the dye molecule and DMF as compared to ACN leads enhanced molecular planarity leading to extended effective π-conjugation, which is supported by a theoretical molecular orbital calculation (MO) calculation using Gaussian (G09) program [49]. A perusal of the optimized molecular structure of the DSQ dye reveals that, unlike typical mono-squaraine dyes exhibiting planar dye molecular structure, double squaraine dye exhibits a slightly non-planar bent structure in the isolated gaseous state. Interaction of the dye with strong HBA solvents leads to the hydrogen bond-assisted molecular planarization, resulting in the extension of the effective π-conjugation and red-shift in the absorption maximum (Appendix A).

### 2.3. Harnessing the Potentiality of DSQ Dye in Metal Ion Sensing

Metal ion sensing for the environment and health monitoring is a fast-growing research field where solvatochromic dye probes are expected to play a dominant role. Using a series of D-π-A molecular framework-based visible light absorbing solvatochromic dyes, Xie et al. demonstrated sensing of Na^+^, K^+^ and H^+^ ions and K^+^ ion selective nanosensor [50,51]. Although their proposed dyes exhibited solvatochromic behavior, the extent of the solvatochromic shift was small and exhibited negative solvatochromism with the solvent polarity. Considering the contrasting absorption spectral features of DSQ dye in ACN and DMF, we dissolved DSQ dye in different combinations of solvent mixtures such as DMF/Water (1:1, *v*/*v*) and DMF/ACN (1:99, *v*/*v*) and explored the differential metal ion sensing behavior.

#### 2.3.1. Investigation of Metal Ion Sensing in the DMF/Water Solvent System

10 μM solution of DSQ dye prepared in the DMF/H_2_O (1:1, *v*/*v*) was treated with the 5 equivalents of different mono and multivalent ions such as Li^+^, Na^+^, K^+^, Mg^2+^, Ca^2+^, Zn^2+^, Mn^2+^, Ba^2+^, Cs^+^, Pb^2+^, Ag^+^, Cd^2+^, Al^3+^, Sr^2+^, Cu^2+^, Fe^3+^, Cr^3+^, Fe^2+^, Ni^2+^, Co^2+^ and obtained results in terms of change color along with the absorption spectral feature are shown in Figure 3. It can be seen from the photographs shown in Figure 3a that blank containing only DSQ dye in this solvent system exhibits cyan color and there was change in the color of the solution in the presence of some metal ions such as Ag^+^, Al^3+^, Cu^2+^, Fe^3+^, Cr^3+^, Fe^2+^, where this changes in color in the presence of Ag^+^ and Cu^2+^ ions were highly conspicuous as visualized by the naked eye. To have an in-depth insight about these visual color changes, absorption spectra solution of the dye in the presence and absence of metals ions were also recorded and shown in Figure 3b. The cyan color of the blank solution is associated with monomeric dye absorption mainly in the NIR wavelength region appearing at the λ_max_ of 706 nm. In the presence of Al^3+^ and Cr^3+^, the color of the solution changed from cyan to the bright blue along with clear change absorption spectral feature, where absorbance at 706 nm was found to decrease and there was the appearance of a new and blue-shifted peak appearing at the λ_max_ of 620 nm with the isosbestic point around 648 nm. In the presence of Fe^3+^, Fe^2+^ visible change in color of the solution was small and is associated with absorption spectral changes with decrease in peak at 706 nm with enhanced absorption at 620 nm and attributed to the hypochromic effect by the Fe^3+^, Fe^2+^ ions to the DSQ dye in DMF/H_2_O (1:1) solvent system. A perusal of Figure 3a indicates an obvious and remarkable change in the color of the DSQ dye (**1**) in the DMF-water (1:1) solvent system from cyan to light yellow in the presence of Cu^2+^ and Ag^+^ ions indicating their very strong interaction with the DSQ dye (**1**). This strong interaction was further confirmed by the clear absorption spectral changes as shown in the Figure 3b exhibiting highly diminished optical absorption around 706 nm. At the same time, with the increasing concentration of these ions, there was a fast and sharp decrease in the absorption peak associated with the monomeric dye molecules appearing at 706 nm, further validating their very strong interaction with the DSQ 1 dye molecules.

This DMF/H_2_O (1:1) solvent system exhibits sensing behavior (although not selective for a particular ion) with the Ag^+^, Al^3+^, Cu^2+^, Fe^3+^, Cr^3+^, Fe^2+^, concentration dependent spectral change with these ions was measured, which is shown in the Figure 3c. It can be seen from this figure that, in the presence of these ions, absorption at 706 nm associated with the monomeric dye molecules decreases as a function of increasing ion concentration this change in the absorbance was more pronounced in the metal ion concentration range of 10^−6^ M–10^−4^ M). Therefore, it can be concluded that the dye DSQ dye exhibits sensing towards Ag^+^, Al^3+^, Cu^2+^, Fe^3+^, Cr^3+^, Fe^2+^ ions of the DMF/H_2_O (1:1) solvent system but considering the strong solvatochromism in this newly designed dye, selectivity for a particular ion could be possible by the judicious selection of a suitable solvent or solvent mixture.

#### 2.3.2. Selective Metal Ion Sensing Utilizing DMF/ACN Solvent System

Our exploration for selective metal ion sensing in different solvents resulted in the selection of DMF/ACN solvents owing to their contrasting spectral changes from visible (ACN) to NIR (DMF) wavelength region graduation change in the spectral features upon changing their relative ratio as shown and discussed in the Section 2.2. To accomplish this, 10 μM solution DSQ dye in the (1:99, *v*/*v*) DMF/ACN solvent system was treated with the five equivalent different metal ions, such as Na^+^, K^+^, Mg^2+^, Ca^2+^, Zn^2+^, Mn^2+^, Ba^2+^, Cs^+^, Pb^2+^, Ag^+^, Cd^2+^, Al^3+^, Sr^2+^, Cu^2+^, Fe^3+^, Cr^3+^, Fe^2+^, Ni^2+^, Co^2+^, and photographs of visual color change along with the absorption spectral behavior in the presence and absence of the different metal ions are shown in Figure 4. A perusal of this figure clearly corroborates that DSQ dye in this solvent system light blue color (blank) and there was a conspicuous change in the color of solution from blue to bright red-violet in the presence of Cu^2+^ ion only as shown in Figure 3a demonstrating the potential of this dye to work as a selective Cu^2+^ ion chemosensor. Although monitoring metal ions play dominant role in controlling the environmental pollution and human health conditions, amongst various metal ions Cu^2+^ is the 3rd most abundant metal ion in the human body, controlling various physiological processes [52]. Increased levels of this ion in the human body leads to several diseases such as Menkes syndrome, Wilson’s disease, Alzheimer’s disease, etc. [53,54,55]. Electronic absorption spectra of the DSQ dye in this solvent system (DMF/ACN, 1:99), as shown in Figure 4b, exhibit λ_max_ at 592 nm corresponding to the observation of the blue color. It is interesting to see that except in the presence of the Cu^2+^ ion, presence of other metal ions does not result in any change in the absorption spectral feature, validating the selective detection of Cu^2+^ ions by the DSQ dye in this solvent system. It can also be seen that after the addition of Cu^2+^ ions, the absorbance of DSQ dye appearing at the 592 nm decreases and a new peak appears at the λ_max_ at the 554 nm, which is responsible for the appearance of the red-violet color. The reason behind this shift peak can be attributed to formation of higher H-aggregation of DSQ-Cu^2+^ complex with increase in concentration of Cu^2+^ ions. On the contrary, after adding the same concentration of other ions, there was almost no change in the absorption spectra further suggesting that DSQ dye in this solvent system could serve a colorimetric and selective Cu^2+^ chemosensor. One can argue that sensitivity of the dye with Cu^2+^ ion is not due to aggregation but associated with complex formation without undergoing the molecular aggregation. He at al has reported the formation of complex on addition of metal ion to their dye, which imparts fluorescence [56]. But, in case of DSQ dye, this possibility is ruled out, since the dye is not showing fluorescence in presence of Cu^2+^. This is due to the formation of strong H-aggregation in the dye ion system. These π-π interactions might well be strongly reflected by the overlaps in π-conjugated system, facilitating the development of excimers and, as a result, emission quenching and the aggregation induced quenching effect as reported by Ma et al. [57] and Huang et al. [58]. In this present work, lack of the observed fluorescence in the presence of Cu^2+^ ions clearly corroborate that the observed behavior is due to aggregation.

In order to further investigate the selective recognition of DSQ dye in the presence of Cu^2+^ ions, concentration dependent titration was conducted. To accomplish this, DSQ dye was dissolved in the DMF/ACN (1:99, *v*/*v*) solvent mixture followed by the addition of different concentrations of Cu^2+^ ions and measurement of the absorption spectra as shown in Figure 5. A perusal of the Figure 5a reveals that addition of Cu^2+^ ions in the 10 μM solution of DMF/ACN (1:99, *v*/*v*) leads to the concomitant decrease in the absorbance at the 592 nm. Therefore, the absorbance at this wavelength was plotted as a function of Cu^2+^ concentration (10^−11^ M–10^−4^ M) as shown in the Figure 5b, which depicts a linear change in the absorbance in the concentration range of nM to μM and after 50 μM, the absorbance becomes constant.

Therefore, the analysis of this Figure 5b reveals that DSQ dye in this solvent system of DMF/ACN (1:99, *v*/*v*) exhibits linear detection of the Cu^2+^ ion in the concentration range 1 nM to 50 μM with the limit of the detection to be 6.5 × 10^−10^ M (Appendix A). It is worth to mention here that World Health Organization have recommended a maximum allowed limit of the Cu^2+^ ions in the drinking water to be up 30 μM [55]. Therefore, present DSQ dye based Cu^2+^ ion sensor can be easily used for checking the quality of the drinking water owing to the visual color change of dye in this solvent system from blue to the red-violet. By the absorption spectrum-based titrimetric investigation, it was found that at lower concentration of Cu^2+^ ions, the DSQ dye-ion binding was guided by a strong electrostatic interactions and quantitative estimation of the binding of Cu^2+^ ions to the dye molecules was analyzed by the Benesi–Hildebrand Equation (1).
(1)$$\frac{1}{A_{0}-A}=\frac{1}{k_{a}\left(A_{1}-A_{0}\right){\left[\mathrm{Cu}2^{+}\right]}^{n}}+\frac{1}{A_{1}-A_{0}}$$
where *A*_o_, *A* and *A*_1_ are the absorbance in absence, intermediate and at maximum presence of Cu^2+^ concentration, respectively, and *k_a_* is the association constant. The plot 1/(*A_o_* − *A*) versus 1/[Cu^2+^]^n^ as shown Figure 5c gives a straight line, when *n* = 2, indicating 2:1 stoichiometry for DSQ-Cu^2+^ complexation. The association constant (*k_a_*) can be determined from the slope of the straight line of this plot, which was found out to be 2.32 × 10^4^ M^−1^ indicating a strong binding of the metal ion with the dye. The value of this k_a_ was further used to calculate Gibb’s free energy change (Δ*G*) for the dye complex formation using the Equation (2).
(2)$$\Delta G=-RT\ln k_{a}$$
where *R* and *T* are universal gas constant and absolute temperature, respectively. The Gibbs free energy change was estimated to be −25.07 kJ mol^−1^ and such a high and negative value of Δ*G* indicates the feasibility of the spontaneous DSQ-Cu^2+^ complex formation. The 2:1 stoichiometry for dye and Cu^2+^ ion complexation estimated by the analysis of the Benesi–Hildebrand equation was further confirmed by using the Job’s plot as shown in Figure 5d. It can be seen from this figure that in the case of [Cu^2+^]/[DSQ-Cu^2+^] to be 0.6, the value of Δ*A* was found to be the largest validating the stoichiometric ratio of DSQ dye and Cu^2+^ to be the 2:1. In practical applications, the sensor must be highly selective and free of interference from other prospective rivals. The less interference there is, the greater the detecting effect. The DSQ dye was analyzed for a matrix of various cations in order to conduct a competitive experimental study (Appendix A). It was discovered that the DSQ dye did not detect the presence of other cations, however on addition of Cu^2+^, it was recognized by the DSQ dye. This signifies that all the cations did not interfere with the identification of Cu^2+^.

## 3. Experimental

### 3.1. General Information

1-iodoethane, diethylsquarate, trimethylamine, thionyl chloride, squaric acid were purchased from Tokyo Chemical Industry Co., Ltd. (Tokyo, Japan) and used as received. Methanol-*d*_4_, DMSO-*d*_6_, CuCl_2_, Ni(NO_3_)_2_, AgNO_3_ were purchased from Sigma-Aldrich (Darmstadt, Germany). 1,1,2-trimethyl-1*H*-benzo[e]indole, acetonitrile, ethanol, NaOH, diethyl ether, toluene, n-Butanol, dimethyl sulfoxide (DMSO), chloroform (CHCl_3_), dichloromethane (DCM), dimethylformamide (DMF), 1,4-dioxane, acetone, acetonitrile (ACN), acetic acid (AcOH), methanol, tetrahydrofuran (THF), ethyl acetate, 2-propanol, deuterated chloroform (CDCl_3_), LiCl, NaCl, KCl, CaCl_2_, MgCl_2_, MnCl_2_·4H_2_O, Ba (NO_3_)_2_, Sr(NO_3_)_2_, Pb(NO_3_)_2_, FeCl_2_·4H_2_O, FeCl_3_, CoCl_2_·6H_2_O, ZnCl_2_, CdCl_2_, AlCl_3_ were purchased from Wako chemical company (Richmond, VA, USA) and used as received. All the aqueous solutions were prepared with Milli-Q water (18.2 MΩ·cm, Millipore S. A. S., France). All reactions were carried out under an atmosphere of N_2_ using anhydrous solvents unless otherwise stated. Synthesized double squaraine dyes and intermediates were confirmed by MALDI-TOF-mass or high-resolution FAB-mass (HR-MS) in positive ion monitoring mode and nuclear magnetic resonance spectroscopy (NMR 500 MHz). The UV-Vis absorption spectrum was obtained with an UV-visible-NIR spectrophotometer (JASCO V-570).

### 3.2. Synthetic Procedure

Synthesis of **3**: To the mixture of 1,1,2-trimethyl-1*H*-benzo[e]indole (6.3 g, 30.0 mmol) in 50 mL acetonitrile was added iodoethane (**7**) (9.6 mL, 120.0 mmol). The resulting mixture was refluxed 18 h. The TLC showed the start material was consumed completely. The reaction was cooled to room temperature, 150 mL ethyl acetate was added, and the resulting precipitate was filtered and washed with ethyl acetate to obtain the grey solid as titled compound **3**, 11.0 g in 100% yield. ^1^H NMR (500 MHz, MeOD) δ 8.33 (d, *J* = 8.5 Hz, 1H), 8.24 (d, *J* = 8.9 Hz, 1H), 8.15 (d, *J* = 8.3 Hz, 1H), 8.04 (d, *J* = 8.9 Hz, 1H), 7.80 (dd, *J* = 6.9, 15.4 Hz, 1H), 7.71 (dd, *J* = 6.9, 15.2 Hz, 1H), 4.70 (q, *J* = 7.4 Hz, 2H), 1.84 (s, 6H), 1.64 (t, *J* = 7.4 Hz, 3H); ^13^C NMR (500 MHz, MeOD) δ 197.06, 139.45, 138.75, 135.15, 132.40, 131.00, 129.66, 129.09, 128.63, 124.41, 113.76, 57.21, 45.18, 22.34, 13.56; HRMS calcd. for C_17_H_20_N^+^ [M^+^] 238.16, found 238.16.

Synthesis of **4**: To the solution of **3** (9.0 g, 24.6 mmol) in 50 mL ethanol was added diethyl squarate (4.2 g, 24.6 mmol) and trimethylamine (10.3 mL, 73.8 mmol). The resulting solution was stirred at room temperature for 15 h giving yellow precipitate. This yellow precipitate was filtered and washed with 100 mL cold ethanol to obtain the titled compound 8.5 g as a yellow solid product in yield 97%. ^1^H NMR (500 MHz, CDCl_3_) δ 8.10 (d, *J* = 8.5 Hz, 1H), 7.88 (d, *J* = 8.2 Hz, 1H), 7.85 (d, *J* = 8.7 Hz, 1H), 7.53 (dd, *J* = 8.2, 15.3 Hz, 1H), 7.38 (dd, *J* = 6.9, 15.0 Hz, 1H), 7.23 (d, *J* = 8.7 Hz, 1H), 5.46 (s, 1H), 4.93 (q, *J* = 7.1 Hz, 2H), 4.01 (q, *J* = 7.2 Hz, 2H), 1.90 (s, 6H), 1.57 (t, *J* = 7.1 Hz, 3H), 1.39 (t, *J* = 7.2 Hz, 3H); ^13^C NMR (500 MHz, CDCl_3_) δ 192.68, 187.09, 186.99, 173.23, 169.85, 139.19, 132.55, 130.77, 129.67, 129.58, 128.54, 127.12, 123.70, 122.13, 109.44, 80.46, 69.83, 49.79, 37.78, 26.55, 15.94, 11.66; TOF mass ESI+ calcd. for C_23_H_23_NO_3_ [M + Na] 384.17, found 384.16.

Synthesis of **5**: To the solution of **4** (1.5 g, 4.2 mmol) in 8 mL ethanol was added 0.8 mL 40% aq. NaOH. The resulting mixture was stirred overnight at room temperature leading to the yellow precipitate. The yellow precipitate was filtered and washed with 10 mL cold ethanol to obtain the titled compound 1.3 g as a yellow solid product at a yield 85%. ^1^H NMR (500 MHz, DMSO-*d*_6_) δ 8.06 (d, *J* = 8.5 Hz, 1H), 7.84 (d, *J* = 8.0 Hz, 1H), 7.81 (d, *J* = 8.7 Hz, 1H), 7.44 (dd, *J* = 8.2, 15.3 Hz, 1H), 7.33 (d, *J* = 8.7 Hz, 1H), 7.23 (dd, *J* = 7.8, 14.9 Hz, 1H), 5.46 (s, 1H), 3.88 (q, *J* = 7.1 Hz, 2H), 1.84 (s, 6H), 1.20 (t, *J* = 7.0 Hz, 3H); ^13^C NMR (500 MHz, DMSO) δ 209.57, 195.17, 178.54, 158.96, 141.03, 129.60, 129.30, 129.22, 129.18, 129.01, 126.72, 121.91, 121.51, 109.73, 83.69, 47.81, 40.00, 36.29, 26.76, 11.25; HRMS calcd. for C_21_H_18_NNaO_3_ [M + H] 356.12, found 356.12.

Synthesis of **6**: To the solution of **3** (3.65 g, 10.0 mmol) in 40 mL dehydrated toluene/*n*-butanol (*v*/*v* = 1/1) was added squaric acid (**9**) (0.57 g, 5.0 mmol). The reaction mixture was subjected to azeotrope reflux at 120 °C using Dean–Stark trap for overnight. After the completion of the reaction, the solvent was evaporated, and the crude product was purified by silica-gel flash column chromatography using chloroform–methanol as an eluting solvent. Collected the pure fraction, evaporated the solvent and dried in vacuum to obtain 2.1 g of the titled compound **6** as a blue black solid in 76% yield. ^1^H NMR (500 MHz, DMSO-*d*_6_) δ 8.23 (d, *J* = 8.5 Hz, 1H), 8.03 (d, *J* = 6.0 Hz, 1H), 8.02 (d, *J* = 5.2 Hz, 1H), 7.71 (d, *J* = 8.9 Hz, 1H), 7.62 (dd, *J* = 6.9, 15.2 Hz, 1H), 7.45 (dd, *J* = 7.8, 14.9 Hz, 1H), 5.86 (s, 1H), 4.27 (q, *J* = 6.9 Hz, 2H), 1.96 (s, 6H), 1.35 (t, *J* = 7.1 Hz, 3H); ^13^C NMR (500 MHz, DMSO-*d_6_*) δ 177.51, 169.81, 139.29, 133.15, 130.84, 129.79, 129.72, 127.96, 127.41, 124.13, 122.21, 111.09, 85.51, 50.52, 38.14, 26.13, 12.06; HRMS calcd. for C_38_H_36_N_2_O_2_ [M + H] 553.28, found 553.28.

Synthesis of **1**:

Method 1: To the suspension of (**5**) (0.36 g, 1.0 mmol) in 10 mL anhydrous ether was added thionyl chloride (0.145 mL, 2.0 mmol) and resulting mixture was stirred for 30 min at 0–5 °C. The solvent and excess thionyl chloride was evaporated mildly under reduced pressure and the residue was dissolved in 12 mL of dehydrated toluene/*n*-butanol (5:1 *v*/*v*) mixture. The reaction mixture was subjected to azeotrope reflux using Dean–Stark trap for overnight. After the completion of the reaction, the solvent was evaporated, and the crude product was purified by silica-gel flash column chromatography using hexane-ethyl acetate as an eluting solvent. Collected the pure purple fraction, dried with Nitrogen gas flow to obtain 65 mg of the titled compound (**1**) as a purple black solid in 10% yield.

Method 2: To the solution of (**6**) (0.55 g, 1.0 mmol) in 12 mL dehydrated toluene/*n*-butanol (*v*/*v* = 5/1) was added squaric acid (**9**) (0.11 g, 1.1 mmol) and the reaction mixture was refluxed at 120 °C for overnight. Squaric acid (**9**) (0.1 g, 1.0 mmol) was added into the above reaction and refluxed for 10 h more. TLC showed the start material was remaining about 30%. Squaric acid (**9**) (0.1 g, 1.0 mmol) was added into the above reaction and refluxed for 8 h more. TLC showed the starting material not reduced any more, therefore, stopped the heating and cooled to room temperature. The solvent was evaporated, and the crude product was purified by silica-gel flash column chromatography using chloroform-methanol as an eluting solvent. Collected the pure blue fraction, evaporated the solvent and dried in vacuum to obtain 160 mg of the titled compound (**1**) as a purple black solid in 25% yield.

^1^H NMR (500 MHz, DMSO-*d*_6_) δ 8.42 (d, *J* = 2.6 Hz, 1H), 8.40 (d, *J* = 3.8 Hz, 1H), 8.31 (d, *J* = 7.2 Hz, 1H), 8.30 (d, *J* = 3.4 Hz, 1H), 8.29 (d, *J* = 1.6 Hz, 1H), 8.07 (d, *J* = 8.46 Hz, 1H), 7.92 (d, *J* = 8.6 Hz, 1H), 7.78–7.84 (m, 2H), 7.53 (d, *J* = 8.9 Hz, 1H), 7.48 (t, *J* = 7.4, 7.8 Hz, 1H), 7.34 (t, *J* = 7.7, 7.2 Hz, 1H), 5.48 (s, 1H), 4.63 (dd, *J* = 7.1, 8.0 Hz, 2H), 4.05 (d, *J* = 7.2 Hz, 2H), 1.78 (s, 6H), 1.74 (s, 6H), 1.46 (t, *J* = 7.2 Hz, 3H), 1.23 (t, *J* = 7.7 Hz, 3H), 1.26 (s, 1H); ^13^C NMR (500 MHz, CDCl_3_) δ 197.31, 194.61, 193.41, 192.65, 180.61, 178.31, 167.91, 167.19, 158.25, 157.59, 139.75, 139.62, 137.30, 133.82, 132.50, 130.97, 130.66, 129.98, 129.72, 129.61, 128.75, 128.52, 128.30, 127.72, 127.05, 123.54, 123.41, 122.10, 112.60, 109.51, 83.91, 82.69, 58.65, 49.78, 45.13, 37.82, 26.45, 26.24, 23.19, 22.73, 13.01, 11.93; HRMS calcd. for C_46_H_36_N_2_O_5_ [M + H] 649.26, found 649.27.

### 3.3. Preparation of Solutions for the Solvatochromism Investigation

Stock solutions were prepared by weighing the solid of DSQ dye on a 5-digital analytical balance and adding ethyl acetate to make a 15 mM solution. Transfer 30 μL of 1.5 mM DSQ dye solution by volumetric pipette into each transparent glass bottle and dried it in vacuum. Then added 3 mL different supper dehydrate solvent into each bottle contained dry DSQ dye and sonicated for 5 min to ensure complete dissolution. The absorption spectra of each sample were measured in duplicate in the wavelength range from 400–800 nm. In the case of compounds **4** and **6**, solvatochromism investigation was conducted by the same method as for the DSQ dye.

### 3.4. Preparation of Stock Solutions of Metal Ions

100 mM metal ion stock solutions were prepared by chloride salt (LiCl, NaCl, KCl, CaCl_2_, MgCl_2_, MnCl_2_·4H_2_O, FeCl_2_·4H_2_O, FeCl_3_, CoCl_2_·6H_2_O, ZnCl_2_, CdCl_2_, AlCl_3_ and nitrate salt (Ba(NO_3_)_2_, Sr(NO_3_)_2_, Pb(NO_3_)_2_ dissolved in Milli-Q water (18.2 MΩ·cm, Millipore). 1 mL of 100 mM each metal ion solution was diluted to 10 mL by adding water to make 10 mM metal ion stock solutions. Different concentration of metal ion solutions were prepared by using similar methods.

### 3.5. Metal Ion Sensing by Absorption Spectral Measurement

All the solutions for measurement were prepared by using small volumes (2–20 μL) of different concentrations of each metal ion, adding into 2 mL DSQ dye solution, and mixing well (20 μL Milli-Q water was added into 2 mL DSQ dye solution, the UV-visible spectrum of the DSQ dye has no effect and it was regarded as a reference sample (blank)). The measurements were taken within 10 min of preparation of solutions. The Cl^−^ anion used in the metal ions (LiCl, NaCl, KCl, CaCl_2_, MgCl_2_, MnCl_2_·4H_2_O, FeCl_2_·4H_2_O, FeCl_3_, CoCl_2_·6H_2_O, ZnCl_2_, CdCl_2_, AlCl_3_) and NO_3_^−^ anion used in the metal ions (Ba (NO_3_)_2_, Sr(NO_3_)_2_, Pb(NO_3_)_2_ have no effect to the DSQ dye in the solvent system of this work.

### 3.6. Density Functional Theory (DFT) Calculation

The structures of structure DSQ dye were optimized using density functional theory with the B3PW91 hybrid functional by using a 6-311 basis set. Theoretical MO computation was performed using the Gaussian 16 program package [49].

## 4. Conclusions

In summary, we have synthesized and characterized a novel and zwitterionic double squaraine dye with unique D-A-A-D π-conjugated molecular structure. Newly proposed double squaraine dye has demonstrated strong positive solvatochromism having absorption spectral changes encompassing from visible to NIR wavelength region. This opens the door for unique and multiple ion sensing using a single probe in judiciously selected solvent systems. The corresponding double squaraine dye exhibited not only large solvatochromic range of about 200 nm but also strong spectral and color changes in the presence of metal ions. Judicious selection of a binary solvent mixture consisted of DMF and ACN (1:99, *v*/*v*), selective detection of Cu^2+^ ion with the linearity from 50 μM to 1 nM have been successfully demonstrated. Benesi–Hildebrand and Jobs plot analysis for interaction between the double squaraine dye and Cu^2+^ ion clearly corroborated the formation of 2:1 dye-Cu^2+^ ion complex with appreciably good association constant of 2.32 × 10^4^ M^−1^. Considering the WHO allowed limit of Cu^2+^ ions intake by human body to be 30 μM, the proposed dye can be used for the simple naked eye colorimetric monitoring of quality of the drinking water.

## Data Availability

The data are available on request.

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
