# Peer review of "Synthesis and Characterization of Newly Designed and Highly Solvatochromic Double Squaraine Dye for Sensitive and Selective Recognition towards Cu^2+^"

_molecules, 2022, doi:10.3390/molecules27196578_

Round 1
Reviewer 1 Report
Please, see the attached file.

Reviewer 2 Report
molecules-1938890-peer-review-v1
The authors in the article "Synthesis and characterization of Newly designed and Highly Solvatochromic Double Squaraine Dye for Metal Ion Sensing" proposed the synthesis and characterization of a novel and zwitterionic double squaraine dye (DSQ) with a unique D-A-A-D structure for Cu2+ ion detection.
The article is interesting and well structured. I consider the manuscript original and ready for publication, because the solvatochromic experiments are well conducted and well presented in terms of figures. The readability is excellent and the discussion is very clear, in my opinion. The tests for metal detection are adequate for publicaton.
Some minor suggestions:
1) In the introduction, some discussion of fluorescent SQ systems are missing, such as the important class of extended DSQ (Maltese et al. Chem. Eur. J. 2016, 22, 10179 – 10186, Corrente et al. Molecules 2021, 26, 6818).
2) Please check the English spelling, typos/errors are seen starting from abstract, as:
Line 16: demonstarted instead of demonstrated.
Line 228 “…at the at the…”
Author Response
Reviewer 2:
Comment summary: The authors in the article "Synthesis and characterization of Newly designed and Highly Solvatochromic Double Squaraine Dye for Metal Ion Sensing" proposed the synthesis and characterization of a novel and zwitterionic double squaraine dye (DSQ) with a unique D-A-A-D structure for Cu2+ ion detection.
The article is interesting and well structured. I consider the manuscript original and ready for publication, because the solvatochromic experiments are well conducted and well presented in terms of figures. The readability is excellent and the discussion is very clear, in my opinion. The tests for metal detection are adequate for publication.
Our Response: We thank the reviewer for his time to review our manuscript, encouragement, helpful comments and valuable suggestions, which are addressed as follows:
Comment 1: In the introduction, some discussion of fluorescent SQ systems are missing, such as the important class of extended DSQ (Maltese et al. Chem. Eur. J. 2016, 22, 10179 – 10186, Corrente et al. Molecules 2021, 26, 6818)
Our Response: We thank the reviewer for this suggestion. We have included this missing information in the revised manuscript as lines 49-50 on the page 2 of the revised manuscript. Also, we have added the suggested references in the revised manuscript as reference [20] and [21].
Comment 2: Please check the English spelling, typos/errors are seen starting from abstract, as:
Line 16: demonstarted instead of demonstrated.
Line 228 “…at the at the…”
Our Response: We thank the reviewer for this suggestion. Suggested typos/errors have been rectified in line 17 of page 1 and line 236 of page 7. Apart from this other such errors found in the manuscript have also been corrected in this revised manuscript.
Reviewer 3 Report
In this manuscript, the authors reported a zwitterionic double squaraine dye DSQ for metal ions sensing. The compound DSQ exhibited strong solvatochromism in the absorption spectra and was able to bind multiple metal ions such as Cu2+, Ag+, Al3+ and Cr3+, which led to different colorimetric responses. Interestingly, when DMF/ACN (1:99, v/v) was used as the solvent, a highly selective and sensitive sensor was achieved for Cu2+. Considering the interesting performance of DSQ, I think it can be considered for publication with minor revisions.
1. Only colorimetric responses were reported, while DSQ could also be fluorescent. The authors should discuss whether DSQ show any fluorescent responses toward metal ions?
2. Closely related literatures should be cited, such as: Anal. Chem., 2021, 93, 14256-14262.; Sens. Actuators B, 2020, 310, 127855.; J. Am. Chem. Soc., 2020, 142, 20306-20312.
Author Response
Reviewer 3:
Comment summary: In this manuscript, the authors reported a zwitterionic double squaraine dye DSQ 1 for metal ions sensing. The compound DSQ exhibited strong solvatochromism in the absorption spectra and was able to bind multiple metal ions such as Cu2+, Ag+, Al3+ and Cr3+, which led to different colorimetric responses. Interestingly, when DMF/ACN (1:99, v/v) was used as the solvent, a highly selective and sensitive sensor was achieved for Cu2+. Considering the interesting performance of DSQ, I think it can be considered for publication with minor revisions.
Our Response: We thank the reviewer for his acceptance and time to review our manuscript, helpful comments and suggestions, which are suitable addressed in the revised manuscript as follows:
Comment 1: Only colorimetric responses were reported, while DSQ could also be fluorescent. The authors should discuss whether DSQ show any fluorescent responses toward metal ions?
Our Response: We agree with this concern of the referee and in fact this concern has also been raised by the referee 1. This compound DSQ was found to be non- fluorescent. Efforts were also made to measure the fluorescence spectra of the dye in different solvents under investigation in the presence and absence of the metal ions. But, we could not observe the fluorescence that is why we restricted our investigations up to absorption spectral studies only.
Comment 2: Closely related literatures should be cited, such as: Anal. Chem., 2021, 93, 14256-14262.; Sens. Actuators B, 2020, 310, 127855.; J. Am. Chem. Soc., 2020, 142, 20306-20312.
Our Response: We thank the reviewer for this suggestion. Agreeing with the referee, suggested references have been added in the revised manuscript as references [45], [46] and [7].
Round 2
Reviewer 1 Report
I thank the authors for carefully considering the suggestions presented. I consider the manuscript to be of excellent standard. Congratulations on the work done.